# The Basis of Tolerance Mechanism to Metsulfuron-Methyl in *Roegneria kamoji* (Triticeae: Poaceae)

**DOI:** 10.3390/plants10091823

**Published:** 2021-09-01

**Authors:** Wei Tang, Shengnan Liu, Xiaoyue Yu, Yongjie Yang, Xiaogang Zhou, Yongliang Lu

**Affiliations:** 1State Key Laboratory of Rice Biology, China National Rice Research Institute, Hangzhou 311400, China; tangwei@caas.cn (W.T.); yuxiaoyue@caas.cn (X.Y.); yangyongjie@caas.cn (Y.Y.); 2Institute of Plant Protection, Sichuan Academy of Agricultural Sciences, Chengdu 610066, China; hnulshn2006@163.com

**Keywords:** common *Roegneria*, ALS inhibitor, non-target site, cytochrome P450, glutathione S-transferases

## Abstract

*Roegneria kamoji*, a perennial monocot weed that belongs to the tribe Triticeae (family: Poaceae), is an emerging problematic weed in winter wheat (*Triticum aestivum*) fields in China. We have previously confirmed four *R. kamoji* populations tolerant to acetyl-CoA carboxylase (ACCase) inhibitors, and failed control of these populations by metsulfuron-methyl was observed. The objective of this study was to characterize the level of tolerance to metsulfuron-methyl, the basis of tolerance mechanism, and cross-tolerance to acetolactate synthase (ALS) inhibitors in *R. kamoji*. A whole-plant dose–response assay showed that plants of all *R. kamoji* populations (both from wheat fields and uncultivated areas) exhibited high tolerance to metsulfuron-methyl, based on their 100% survival at 6-fold recommended field dose (RFD) and ED_50_ values >6.84-fold RFD, no susceptible population was found. Gene sequencing indicated that no reported amino acid substitutions associated with resistance to ALS inhibitor were found in the *ALS* gene among the *R. kamoji* populations. Pretreatment with the known cytochrome P450 monooxygenases (CytP450) inhibitor malathion reduced the ED_50_ values of metsulfuron-methyl in two *R. kamoji* populations. These populations also exhibited cross-tolerance to RFD of mesosulfuron-methyl and bispyribac-sodium. The activities of glutathione-S-transferase (GST) and CytP450 could be induced by metsulfuron-methyl in *R. kamoji*, which is similar to the known tolerant crop wheat. This is the first report elucidating metsulfuron-methyl tolerance in *R. kamoji*. The reversal of tolerance by malathion and the GST and/or CytP450 enhanced herbicide metabolism suggests that non-target-site mechanisms confer tolerance to metsulfuron-methyl in *R. kamoji*.

## 1. Introduction

*Roegneria kamoji* is a common perennial weed that belongs to *Roegneria* of the tribe Triticeae (Poaceae family). It is widely distributed across China, Korea, and Japan, and is usually found in hillside, grassland, urban green spaces, and field borders [1,2]. In recent years, *R. kamoji* has been found spreading in winter wheat (*Triticum aestivum*) fields in Hubei, Anhui, and Zhejiang provinces of China (Appendix A), and has become a dominant weed in some regions [3,4].

It has been the practice of many farmers to use acetyl-CoA carboxylases (EC 6.4.1.2, ACCase) inhibitors, such as fenoxaprop-ethyl, clodinafop-propargyl, and pinoxaden for postemergence control of graminaceous weeds in wheat [5,6]. Failed control of *R. kamoji* by fenoxaprop-ethyl was observed in both populations collected from wheat fields and uncultivated areas, which implies there was no selection or genetic manipulation to make this weed tolerant; it is naturally tolerant. The tolerance mechanism was due to non-target mutations and an enhanced ACCase activity after herbicide treatment [3]. Once ACCase inhibitor tolerance was observed, growers will generally start to use acetolactate synthase (EC 4.1.3.18, ALS) inhibitors as an alternative for control of ACCase resistant weeds. Metsulfuron-methyl has been one of the most important ALS inhibitors used for grass weed control in wheat [7,8]. Unfortunately, poor control efficacy of metsulfuron-methyl has been observed for these ACCase inhibitor-tolerant *R. kamoji* populations in a preliminary screening (Appendix A).

ALS inhibitors, which inhibit the activity of the enzyme ALS that catalyzes the first reaction in the biosynthesis of branched-chain amino acids (isoleucine, leucine, and valine), can be separated into five classes: sulfonylurea (SU), imidazolinone (IMI), sulfonylaminocarbonyl-triazolinones (SCT), triazolopyrimidine (TP), and pyrimidinyl thiobenzoate (PTB) based on the chemical structures [9,10,11]. Currently, resistance/tolerance to ALS inhibitors is very common worldwide—167 weed species (65 monocots and 102 dicots) have been documented with resistance to ALS inhibitors, accounting for one-third of the total reported resistant cases [12]. In most cases, target-site resistance (TSR) caused by point mutations resulting in single amino acid substitutions in the *ALS* gene is mainly responsible for resistance to ALS inhibitors. To date, at least 29 amino acid substitutions have been identified at eight sites [13,14,15,16,17]. However, the non-target-site resistance (NTSR) mechanism, endowed by the metabolism of ALS inhibitors by key enzymatic complexes such as glutathione *S*-transferases (GST) and cytochrome P450 monooxygenases (CytP450), was also identified in some weed species [18,19,20,21].

Selective mechanism of ALS inhibitors may occur due to differential rate of absorption, translocation, sequestration, and deactivation between weed species and wheat [22,23]. Weed species in the same tribe of wheat are structurally similar or genetically related, they may share similar response patterns to a specific stress [24]. For example, for *Aegilops tauschii*, an annual weed of the tribe Triticeae, effective herbicide options become limited due to its phylogenetic closeness to wheat [25,26,27]. It is reported that mesosulfuron-methyl is the only wheat-registered foliar-applied herbicide that provides control of *A. tauschii* in China [27].

*R. kamoji* is genetically similar and has a parallel life cycle and growth habits with wheat [28], very little information is currently available regarding the response of this weed to ALS inhibitors. Therefore, the objectives of this study were to: (1) determine the tolerance level and the basis of tolerance mechanism to metsulfuron-methyl in *R. kamoji*, and (2) to determine the cross-tolerance to a single dose of other classes of ALS inhibitors in *R. kamoji*.

## 2. Results

### 2.1. Dose-Response to Metsulfuron-Methyl

The dose–response experiments indicated that all *R. kamoji* populations showed similar response patterns with the increasing metsulfuron-methyl dose, and all plants survived from the treatment of metsulfuron-methyl at 45 g ai ha^−1^ (6-fold recommended field dose (RFD), Appendix A). As shown in Table 1, the effective dose for 50% fresh weight reduction (ED_50_) value of the four *R. kamoji* populations was over 50 g ai ha^−1^ and 6.8-fold greater than that of the RFD dose of metsulfuron-methyl. These results suggested that *R. kamoji* had high tolerance levels to metsulfuron-methyl.

### 2.2. Effect of Malathion on Metsulfuron-Methyl Tolerance

As a result of no differences among the four *R. kamoji* populations in their response to metsulfuron-methyl, HBJZ, and ZJHZ populations were selected to investigate the effect of malathion on metsulfuron-methyl tolerance. When malathion was applied alone, no obvious effect on plant growth was observed, and no influence on the above-ground biomass was detected in either HBJZ or ZJHZ population (Figure 1). However, under malathion pretreatment, the metsulfuron-methyl ED_50_ values decreased 46% and 64%, from 51.3 to 27.8 and 55.1 to 20.1 for HBJZ and ZJHZ populations, respectively (Figure 2). This finding suggested that CytP450s likely contribute to metsulfuron-methyl tolerance in *R. kamoji*.

### 2.3. ALS Gene Amplification and Sequencing

After BLAST analysis of the ALS amino acid of *R. kamoji* (GenBank accession MZ368697) in the NCBI database, we found that the ALS amino acid of *R. kamoji* has 99% identity to wheat (*Triticum aestivum*) and 73% identity to *Arabidopsis thaliana* (Figure 3). Using BioEdit to compare the amino acid sequence of four *R. kamoji* populations, *A. thaliana*, and *T. aestivum*, the results showed that some amino acids of *R. kamoji* are inconsistent with *T. aestivum*, but none of them were related to the reported resistance-associated substitutions. These results indicated that the tolerance to ACCase inhibitors in *R. kamoji* populations may be caused by non-target-site tolerance mechanisms.

### 2.4. Enzyme-Linked Immunosorbent Assay (ELISA) of ALS, CytP450 and GST Activities

The enzyme ELISA tests over a period of 14 d indicated that activities of ALS, CytP450, and GST in *R. kamoji* ZJHZ were close to that of *T. aestivum*, and showed similar responses after metsulfuron-methyl treatment. ALS activity decreased in both *R. kamoji* and *T. aestivum* plants, and reached a minimum at 7 days after treatment (DAT), then gradually increased to 58% and 62% of the 0 DAT vales at 14 DAT, respectively (Figure 4). However, the CytP450 and GST activities could be induced by metsulfuron-methyl for both *R. kamoji* and *T. aestivum*. After metsulfuron-methyl treatment, CytP450 activity increased and peaking at 3 DAT, then decreased and maintained equivalent or greater activities from 7 to 14 DAT for both *R. kamoji* and *T. aestivum*. These results indicated that the target enzyme (ALS) activity was not the main reason for herbicide tolerance in *R. kamoji*, the induced increase in CytP450 and GST activities provide evidence that a non-target-site mechanism, probably via CytP450 and/or GST-mediated detoxification of the herbicide, is likely conferring tolerance to metsulfuron-methyl in *R. kamoji* plants.

### 2.5. Single-Dose ALS Herbicides Cross-Tolerance Testing

This study found that the response of ZJHZ and HBJZ *R. kamoji* populations to ALS herbicides at their RFD varied depending on herbicide classes (Table 2). Both ZJHZ and HBJZ plants were susceptible (no survival plants and <15% fresh weight of control) to flucarbazone-sodium, imazapic, and pyroxsulam, while all *R. kamoji* plants showed moderate tolerance (100% survival and >45% fresh weight of control) to mesosulfuron-methyl and bispyribac-sodium. The ED_50_ values of ZJHZ and HBJZ to mesosulfuron-methyl were also 1-fold greater than that of the RFD dose, and there was a significant reduction in mesosulfuron-methyl tolerance in the presence of malathion for the two *R. kamoji* populations (Appendix A). These results indicated that *R. kamoji* also exhibited cross-tolerance to SU and PTB families of ALS herbicides.

## 3. Discussion

Metsulfuron-methyl is widely known for its low use doses, high efficacy and crop selectivity, and broad-spectrum in controlling many broadleaf and grass weeds [29]. Resistance to Metsulfuron-methyl has been reported in several monocotyledonous weeds, such as *Lolium rigidum* [21,30], *Avena fatua* [12], and *Polypogon fugax* [31]. In this study, the four *R, kamoji* populations showed no symptoms after being treated with metsulfuron-methyl at recommended field dose. In comparison, the survival of a susceptible *Raphanus sativus* was reduced by more than 99% with only 1/5 of the commercial field rate (6 g ai ha^−1^) [32]. In another whole-plant dose–response study, ED_50_ values of *Eclipta prostrata* and *P. fugax* to metsulfuron-methyl were 0.07 and 8.57 for the S population, respectively [11,31]. From this point, *R. kamoji* populations were highly tolerant to metsulfuron-methyl.

These results from malathion plus metsulfuron-methyl application experiments are in accordance with studies conducted in other weed species such as *Amaranthus palmeri* [13], *Myosoton aquaticum* [14], and *A. tuberculatus* [33]. However, there are over 5100 sequences of plant CytP450 that have been annotated and named, and each *CytP450* gene participates in various biochemical pathways to produce primary and secondary metabolites [34]. To further investigate the mechanisms of metsulfuron-methyl tolerance, the transcriptome analysis of *R. kamoji* populations under herbicide treatment is currently in progress in our laboratory to identify candidate CytP450 genes involved in metsulfuron-methyl tolerance.

The differential sensitivity among populations might be due to inherent genetic variation and also due to environmental adaptations [23]. To investigate the tolerance mechanism of *R. kamoji* populations to metsulfuron-methyl, the target *ALS* gene was isolated from the four *R. kamoji* populations. To our knowledge, this is the first report regarding the full-length *ALS* gene in *R. kamoji*. Both populations from wheat fields and uncultivated areas share a similar sequence, which is also close to the *ALS* gene of the known tolerant crop wheat. This result is in accordance with the malathion pretreatment experiment, suggesting that tolerance to metsulfuron-methyl in *R. kamoji* is not caused by the target site mechanism.

CytP450 are heme-containing monooxygenases involved in both biosynthetic and detoxification pathways in many plants [35,36]. It is reported that ALS inhibitors, such as chlorotoluron in wheat and barley, and pyrazosulfuron-ethyl in rice are metabolized by CytP450s [37,38]. Malathion is a known CytP450 inhibitor, which will bind the enzyme that is detoxifying the herbicide [38]. In this study, malathion was used as an indicator for detecting metabolic tolerance to metsulfuron-methyl, and reduced CytP450 metabolism of metsulfuron-methyl was observed. These results are in agreement with those for other weed species such as *Myosoton aquaticum* [14], *A. tauschii* [25], and *P. fufax* [32]. GST also plays an important role in resistance to particular ALS inhibitors in some weed species [14,32]. In wheat, herbicide safeners, such as cloquintocet mexyl, mefenpyr diethyl can induce GST activity, thereby reducing injury to ACCase inhibitors [39]. Our results indicated that ALS activity was inhibited from 0 to 7 DAT after being treated with metsulfuron-methy, increased activities of GST and CytP450 from 0 to 5 DAT are likely to promote the metabolism of metsulfuron-methy and confer tolerance to this herbicide in *R. kamoji*.

Weed species segregating NTSR mechanism often confers unpredictable cross-resistance patterns to herbicides of other classes in the same chemical family [40]. For instance, a resistant *A. tauschii* population with enhanced mesosulfuron-methyl metabolism was also resistant to IMI and TP herbicides, but susceptible to PTB herbicide in ALS inhibitors [25]. However, in this study, *R. kamoji* populations were also tolerant to SU and PTB herbicides but susceptible to IMI, TP, and SCT herbicides in ALS inhibitors. These results would be helpful for farmers in developing more effective herbicide application programs for managing this weed.

In summary, this is the first report to confirm metsulfuron-methyl tolerance and cross-tolerance to ALS inhibitors in *R. kamoji* populations. The basis of tolerance to metsulfuron-methyl was conferred by a non-target-site mechanism, likely enhanced the detoxification of the herbicide, playing a crucial role in exhibiting tolerance. More importantly, the close phylogenic relationship between *R. kamoji* and *T. aestivum*, combined with high seed production and efficient seed and rhizome dispersal [3,28], might become a challenge in many cropping systems. Farmers should be encouraged to use herbicides with different modes of action, as well as adopt sustainable and effective weed management strategies to control this weed.

## 4. Materials and Methods

### 4.1. Plant Materials and Growth Conditions

Seeds of four *R. kamoji* populations were used in this study, including two populations collected from wheat fields (HBJZ and ZJJX) where failed control by fenoxaprop-ethyl were observed, and two populations from non-cultivated areas (HNHY and ZJFY). Details of these populations can be found in our previous studies [3]. In a preliminary experiment, seedlings of these *R. kamoji* populations survived at 4-fold recommended field dose (RFD), no susceptible *R. kamoji* population was determined (data not shown). A wheat cultivar (Yangmai 25) was used as an ALS-inhibitor-tolerant standard for ALS, GST, CytP450 enzyme activities comparison with *R. kamoji* after metsulfuron-methyl treatment in this study.

Seeds for all experiments were germinated in plastic trays (28 cm × 18 cm × 7.5 cm) containing a double layer of moistened filter paper (Double Ring #102, Hangzhou Special Paper Industry Co. Ltd., Hangzhou, China) at 25/15 °C with 14 h light coinciding with the high-temperature period. Germinated seeds with 2 mm emerged radicle were transplanted into 9-cm-diameter plastic pots containing potting soil (Hangzhou Jin Hai Agriculture Co., Ltd., Hangzhou, China). The pots were placed in a screenhouse (a 6 × 40-m chamber framed with 2-cm iron mesh and covered overhead with a transparent plastic cover to prevent rain damage, about 25/15 °C, natural light) at the China National Rice Research Institute (CNRRI, 30°04′ N, 119°55′ E) and watered as required to maintain soil moisture. There were four uniform seedlings in each pot grown to three- to four-leaf stage for herbicide spraying.

### 4.2. Dose Response to Metsulfuron-Methyl

*Roegneria kamoji* seedlings at the 3-4 leaf stage were sprayed with metsulfuron-methyl (Table 3) at 0, 1/2-, 1-, 1.5-, 3-, 6-, 12-, 24, and 48-fold of the RFD (7.5 g ai ha^−1^). Herbicides were applied using a laboratory cabinet sprayer (3WP-2000, Nanjing Institute of Agricultural Mechanization Ministry of Agriculture, Nanjing, China) equipped with a flat-fan nozzle (TP6501E) to deliver 200 L^−1^ at 230 kPa. Plants were returned back to the screenhouse and the pots were arranged in a randomized complete block design. At 21 DAT, the above-ground shoot biomass was harvested and the fresh weight was recorded. Four pot replicates were used for each herbicide treatment, and the experiment was repeated once under similar conditions.

### 4.3. Effect of Malathion on Metsulfuron-Methyl Tolerance

Malathion is an organophosphate insecticide and acaricide that has been used as an indicator of CytP450 involvement in metabolic resistance to ALS herbicides [14,25]. The response of HBJZ and ZJHZ populations to metsulfuron-methyl plus malathion was evaluated. Plants were treated with 0 or 1000 g ai ha^−1^ malathion 1 h prior to the application of metsulfuron-methyl with different rates as described above. Non-treated seedlings and seedlings treated only with malathion were used as respective controls to compare the efficacy of malathion in changing the sensitivity of the *R. kamoji* plants to metsulfuron-methyl. Assessments were carried out at 21 DAT as described above.

### 4.4. ALS Gene Amplification and Sequencing

To investigate whether mutations in the ALS gene contributed to the metsufuron-methyl tolerance, fresh leaf tissue (100 mg) was collected from plants of the four *R. kamoji* populations (ten individuals per population) that survived from metsulfuron-methyl treatments in the dose-response experiments. The collected tissue samples were frozen in liquid nitrogen, and total DNA was extracted by using the Plant Genomic DNA Kit (Tiangen Biotech, Beijing, China), following the manufacturer’s instructions. A pair of primers (ALSF: 5′-CTCGCCCGTCATCACCAA-3′ and ALSR: 5′-TCCTGCCATCACCCTCCA-3′) were designed to amplify the ALS gene of ~1600 bp containing the eight known resistance-conferring mutation sites, and the PCR protocols have been described elsewhere [31]. The PCR products were detected with 1% agarose gel and purified using the TIANgel Midi Purification Kit (Tiangen Biotech, Beijing, China). The purified product was sequenced using the ALSF and ALSR primers with the Sanger method by a commercial corporation (Biosune Biotechnology Co., Ltd., Shanghai, China). Alignment and comparison of the sequence data were performed using BioEdit software (Version 7.2.5).

### 4.5. Enzyme-Linked Immunosorbent Assay (ELISA) of ALS, CYP450 and GST Activities

To determine whether the tolerance in *R. kamoji* is caused by the insensitive target enzyme or enhanced metabolic enzyme, activities of ALS, CytP450, and GST toward metsulfuron-methyl for the untreated and treated plants of the ZJHZ population was analyzed and compared with *T. aestivum* over a period of 14 d. Seedlings of both *R. kamoji* ZJHZ and wheat were cultivated to the three-leaf stage as described above. Seedlings were sprayed with metsulfuron-methyl at 45 g ai ha^−1^ and 2 g fresh leaf tissue was collected at 0, 1, 2, 3, 5, 7, 9, 11, and 14 DAT. The leaf tissue was treated with PBS prior to biochemical assays after ground with liquid nitrogen. A fresh leaf sample (0.1 g) was homogenized by 0.9 mL of PBS at pH 7.2–7.4 and centrifuged at 3500 rpm for 15 min at 4 °C. The supernatant was collected in a centrifuge tube and placed in an ice bath. Activities of ALS, GST, and P450 were determined by using ELISA kits (Meimian Biotechnology Co., Ltd., Yancheng, China) according to the manufacturer’s instructions. Each treatment included four replications, and the experiment was repeated once.

### 4.6. Single-Dose ALS Herbicides Cross-Tolerance Testing

The HBJZ and ZJHZ populations of *R. kamoji* seedlings were planted and grown under the screenhouse as described above. In order to investigate the cross-tolerance of *R. kamoji* to other classes of ALS herbicides, seedlings of HBJZ and ZJHZ at the 3~4 leaf stage were treated separately with labeled field recommended rates of mesosulfuron-methyl; imazapic, pyroxsulam, and bispyribac-sodium, which belong to the SU, IMI, TP, and PTB classes of ALS herbicides, respectively. Details of the herbicides are listed in Table 3. The method of herbicide spraying was described earlier. Plant above-ground fresh weight was measured at 21 DAT and the data was expressed as a percentage of the untreated control. We defined a fresh weight percentage >80% of control as high tolerant, 80–20% of control as low tolerant, and <20% of control as susceptible.

### 4.7. Statistical Analysis

The above-ground fresh weight data obtained from the whole-plant dose–response experiment and cross-tolerance experiment were presented as the percentage of untreated control, and subject to ANOVA in SPSS software (v. 13.0, SPSS, Chicago, IL, USA) to test for treatment and experiment interaction. The data of the repeated experiments were pooled, because the interaction of herbicide treatment and experiment was not significant (*p* > 0.05), and then fitted to nonlinear regression analysis in Origin software (v. 2021b, OriginLab Corp., Northampton, MA, USA). The ED_50_ values (herbicide dose required to cause 50% reduction of plant fresh weight) were determined with the use of the following four parameter log-logistic curve [41]:$$y=C+\frac{D-C}{1+{\left(x/ED_{50}\right)}^{b}}$$
where *C* is the lower limit, *D* is the upper limit, *b* is the slope of the curve through ED_50_, *x* is the herbicide dose, and *y* represents plant fresh weight as a percentage of the control. Because no standard susceptible population was available in our preliminary screening experiment, the tolerance level was indicated by calculating the ratio of ED_50_ of the four *R. kamoji* populations and the recommended field dose of metsulfuron-methyl.

## Figures and Tables

**Figure 1 plants-10-01823-f001:**
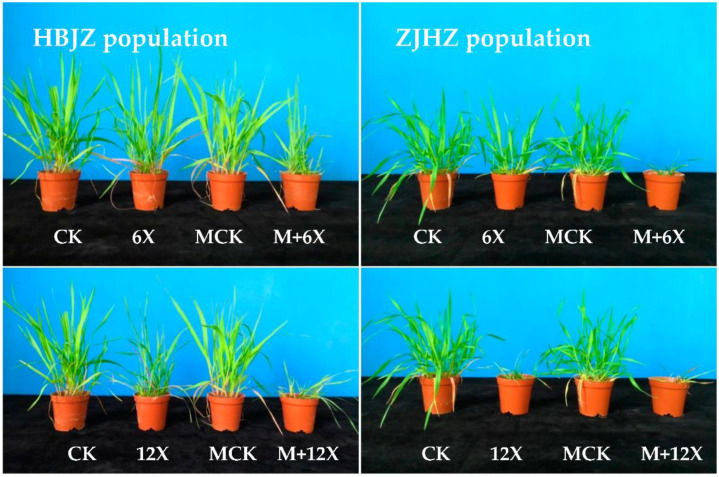
Photographs of *R. kamoji* HBJZ (**left**) and ZJHZ (**right**) populations 21 days after treatment. The first row, showing the untreated control (CK), the metsulfuron-methyl treatment (X represents the recommended field dose of metsulfuron-methyl 7.5 g ai ha^−1^, and 6X, 45 g ai ha^−1^), the malathion treatment control (MCK), the malathion plus metsulfuron-methyl treatment (M + 6X); and the second row, showing the untreated control (CK), the metsulfuron-methyl treatment (12X, 90 g ai ha^−1^), the malathion treatment control (MCK), the malathion plus metsulfuron-methyl treatment (M + 12X).

**Figure 2 plants-10-01823-f002:**
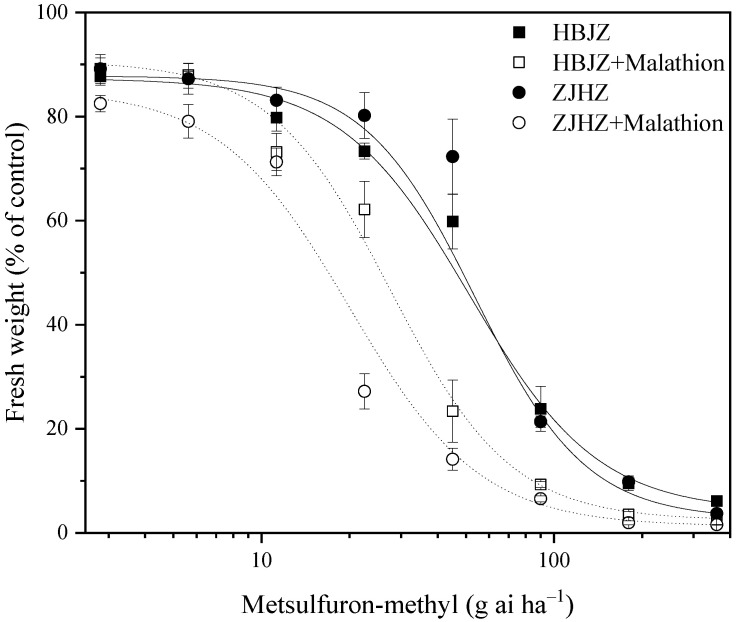
Dose–response curve for the fresh weight (% of control) of the HBJZ and ZJHZ *R. kamoji* populations treated with different doses of metsulfuron-methyl with or without malathion pretreatment. Each point is the mean ± SE of twice-repeated experiments, each including four replicates.

**Figure 3 plants-10-01823-f003:**
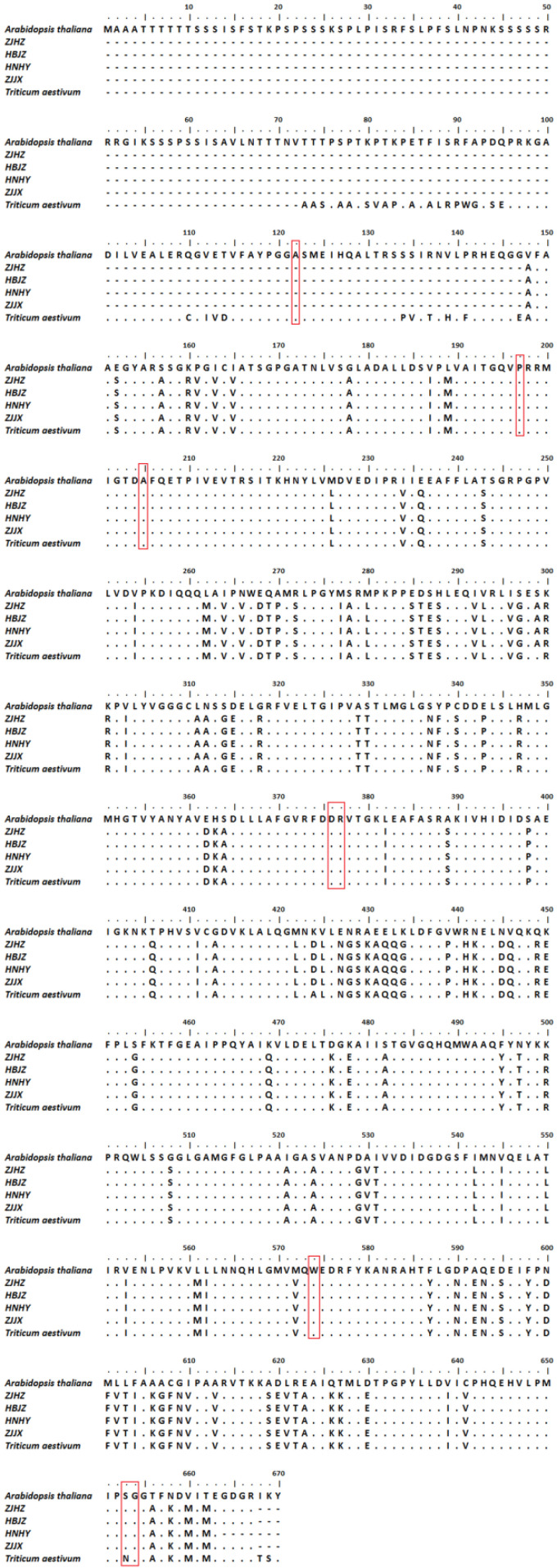
Sequence alignment and analysis of partial ALS gene from four *R. kamoji* populations, *Arabidopsis thaliana* and *Triticum aestivum*. Amino acid numbering refers to the *A. thaliana* ALS gene sequence. The boxed region indicates the eight reported mutations Ala122, Pro197, Ala205, Asp376, Arg377, Trp574, Ser653, and Gly654, which confer target-site resistance to ALS herbicides.

**Figure 4 plants-10-01823-f004:**
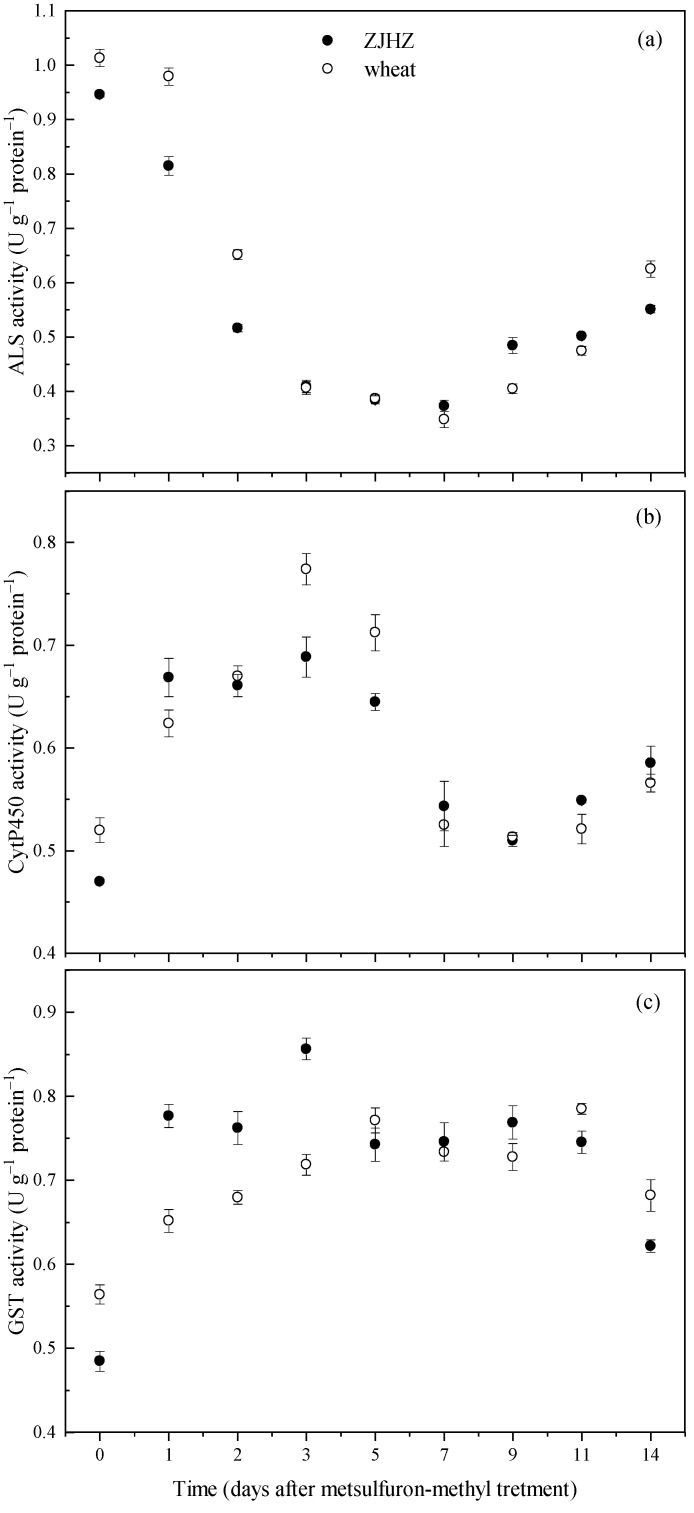
Activities of ALS (**a**), CytP450 (**b**), and GST (**c**) in *R. kamoji* population ZJHZ and compared with *T. aestivum* at 0 to 14 days after metsulfuron-methyl treatment. Each point is the mean ± SE of twice-repeated experiments, each containing four replicates.

**Table 1 plants-10-01823-t001:** The herbicide dose required for 50% fresh weight reduction (ED_50_) and the ED_50_/recommended field dose (RFD, 7.5 g ai ha^−1^) values for metsulfuron-methyl in *R. kamoji* populations.

Population	ED_50_ (g ai ha^−1^) (SE)	ED_50_/RFD
HBJZ	51.3 (4.6)	6.8
HNHY	52.8 (2.7)	7.0
ZJJX	53.3 (3.1)	7.1
ZJHZ	55.1 (4.9)	7.4

**Table 2 plants-10-01823-t002:** Survival percentage (%) and above-ground fresh weight reduction (%) of the HBJZ and ZJHZ *R. kamoji* populations 21 days after treatment with different ALS herbicides.

Herbicide	Survival Percentage (%)		Above Ground Fresh Weight (% of Control)	
	HBJZ	ZJHZ	HBJZ	ZJHZ
Mesosulfuron-methyl	100	100	48.8 (4.9)	47.7 (2.7)
Imazapic	0	0	4.8 (1.2)	90.7 (0.9)
Pyroxsulam	0	0	5.2 (0.6)	91.7 (0.8)
Flucarbazone-sodium	0	0	8.9 (1.2)	14.0 (1.9)
Bispyribac-sodium	100	100	45.3 (0.8)	46.7 (4.3)

**Table 3 plants-10-01823-t003:** Detailed information of ALS herbicides used in this study.

Herbicide	Classes	Formulation and Manufacturer	Recommeded Field Dose (g ai ha^−1^)
Metsulfuron-methyl	SU	10% WP, Jiangsu Tianrong Group, Nanjing, China	7.5
Mesosulfuron-methyl	SU	30 g L^−1^ OD, Bayer, Hangzhou, China	11.25
Imazapic	IMI	240 g L^−1^ AS, BASF, Shanghai, China	144
Pyroxsulam	TP	7.5% WDG, Dow AgroScience, Beijing, China	12
Flucarbazone-sodium	SCT	70% WDG, Arysta LifeScience, Shanghai, China	31.5
Bispyribac-sodium	PTB	10% SC, Kumiai Chemical, Nanjing, China	30

## Data Availability

The data from *ALS* gene sequencing has been submitted into the NCBI database, and the accession number is MZ368697 (https://www.ncbi.nlm.nih.gov/nuccore/MZ368697, accessed on 4 June 2021).

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
