# Peer review of "The Basis of Tolerance Mechanism to Metsulfuron-Methyl in Roegneria kamoji (Triticeae: Poaceae)"

_plants, 2021, doi:10.3390/plants10091823_

Round 1

Reviewer 1 Report

The present study investigated mechanisms of tolerance to select ACCase inhibiting herbicides in four Roegneria kamoji populations. The study contains important results, is well conducted and written. My only question concerns why all four populations were not used in all experiments? In particular, it would be important to know if R. kamoji populations from non-cultivated areas display cross-resistance to ALS inhibitors as this would have important management considerations wherever this species occurs.

Similarly, why is data for all four populations not displayed in all figures and tables?

Reviewer 2 Report

In the manuscript plants-1346885, Tang et al. explore and characterize the mechanisms of cross-tolerance to sulfonylurea and pyrimidinyl thiobenzoate, chemical families of the ALS inhibiting herbicides, in the grassweed Roegneria kamoji, a weed that has become common in wheat cultivation in China and that has already been reported with tolerance to ACCase inhibitors.
The experimental work was well conducted, following standardized methodologies in Weed Science. The results are well described and discussed, and in general, I have not found any major flaws, with the exception of typographical errors throughout the text. My main suggestions are the follows:
1. I recommend that the authors describe the difference between resistance and tolerance to herbicides in the Introducction
2. Show the tables and figures immediately after the paragraph that were mentioned.
3. Rectify the proper order of tables and figures
4. In the description of some results, the authors quickly summarize the methodology, which is unnecessary. I suggest to remove the statements about methodology in the results section.
5. In the discussion the authors should emphasize that Roegneria kamoji is the first species found with multiple (natural) tolerance to ALS and ACCase inhibitors. This information is vaguely mentioned in the abstract and in the introduction, but is not clearly reiterated in the study's conclusion.
Other minor comments or typographical errors are marked in the attached PDF

Reviewer 3 Report

The review concerns the manuscript entitled “The basis of tolerance mechanism to metsulfuron- methyl in Roegneria kamoji (Triticeae: Poaceae)”.  The aims of study were to: determine the tolerance level and the basis of tolerance mechanism to metsulfuron-methyl in R. kamoji, and to determine the cross-tolerance pattern to other classes of ALS inhibitors in R. kamoji.

The manuscript is interesting, but needs to improve.

In my opinion, the second part of objective in the manuscript has not been achieved (“to determine the cross-tolerance pattern to other classes of ALS inhibitors in R. kamoji”). Please correct this part of the research objective.

According to "Materials and methods" chapter the plants assessments were carried out at 21 days after treatment, but figures and tables show the data  after 28 days after treatment. Please revise it.

In "Materials and methods" chapter please describe how the ED50 was calculated.

Please correct the captions in Figure 1.

In the caption of chapter 2  should be "Results", not "Results and Discussion".

The Discussion chapter should be improved. Please add information on metsulfuron-methyl resistance of other monocotyledonous weeds, for example Avena fatua and Lolium sp. Please explain why malathion reduces weed tolerance to metsulfuron-methyl.

Please verify the following sentence: “Our results indicated that ALS activity was 201 significantly inhibited after treated with metsulfuron-methy, induced activities of total GST 202 and CytP450 are responsible for tolerance to metsulfuron-methy in R. kamoji.”

The manuscript must be linguistically corrected.

Best regards

Round 2

Reviewer 3 Report

The manuscript was greatly improved. I accept the responses to my comments.

Best regards